Journal of
open psychology data

# Norms on the Gender Perception of Role Nouns: Gender Ratio Data for Chinese, Finnish, and Russian

**DATA PAPER**

**UTE GABRIEL** (iD)

**JONATHAN KIM** (iD)

**ANTON ÖTTL** (iD)

**PASCAL GYGAX** (iD)

**LEI CUI** (iD)

**JUKKA HYÖNÄ** (iD)

**OLGA NAGEL** (iD)

*Author affiliations can be found in the back matter of this article

]u[ubiquity press

## ABSTRACT

The perceived gender ratio of 422 role nouns was evaluated by Chinese- ($N = 80$), Finnish- ($N = 77$), and Russian-speaking ($N = 134$) students using an 11-point rating scale with counterbalanced scale anchors. Data were collected online between 2015 and 2019, via a self-administered questionnaire. The dataset contains all role nouns in English together with their Chinese, Finnish, and Russian translations, as well as by-item analyses. The data set expands previous data sets and provides social scientists with gender ratio information when selecting social or occupational roles as stimulus materials for experimentation, especially for cross-linguistic or cross-cultural studies.

**CORRESPONDING AUTHOR:**

**Ute Gabriel**

Norwegian University of Science and Technology, NO

ute.gabriel@ntnu.no

**KEYWORDS:**
Gender ratio; role nouns; norms; gender associations; gender stereotypes

**TO CITE THIS ARTICLE:**
Gabriel, U., Kim, J., Öttl, A., Gygax, P., Cui, L., Hyönä, J., & Nagel, O. (2023). Norms on the Gender Perception of Role Nouns: Gender Ratio Data for Chinese, Finnish, and Russian. *Journal of Open Psychology Data,* 11: 2, pp. 1–9. DOI: https://doi.org/10.5334/jopd.73

# (1) BACKGROUND

Role nouns are nouns that describe the activities, functions, occupations, and/or positions that a person performs or holds. Role nouns are widely used as stimulus material in social and cognitive psychological and psycholinguistic research, especially in research on the activation and processing of gender information both on the word (e.g., Kim et al., 2022) and sentence level (e.g., Fukumura et al., 2013; Garnham & Yakovlev, 2015; Pyykkönen et al., 2010; for a recent review, see Esaulova & von Stockhausen, 2022).

Gender is one of the primary features in person perception, and research has found that human referents are immediately and effortlessly clustered by gender, even when this has no informational benefits (Ellemers, 2018). With this in mind, some role nouns are semantically linked to a gender (e.g., aunt/uncle; queen/king), while for others, the gender of the person referred to is marked by a lexeme (e.g., chairwoman/chairman). Further, in languages with sex-based grammatical gender systems, the gender of the referent can also be assigned with grammatical or morphological markers, such as suffixes in this example from Russian: *uchitelnitsa* (teacher$_{FEM}$) vs. *uchitel* (teacher$_{MASC}$).

Role nouns do not just carry gender information through lexical and morphological elements but can also be conceptually associated with a gender. The perceived gender ratios of roles are considered to reflect these mental associations, which have (among others) been referred to as the role nouns' conceptual gender or gender stereotypicality (see Kim et al., 2021, for a discussion of the conceptualisation of gender ratio in psychological research).

As role nouns' gender associations (at least partially) reflect observations related to each role (Koenig & Eagly, 2014), cultural variation and changes in line with societal change can be expected and have been reported (e.g., Kennison & Trofe, 2003; Munoz Sastre et al., 2000). As such, perceived gender ratios serve as a basis for the selection of stimulus material (e.g., Esaulova et al., 2014; Irmen, 2007; Wolfram & Mohr, 2010) and are also the subject of research themselves (e.g., Gabriel et al., 2008; Munoz Sastre et al., 2000).

For psychological research, the *perceived* gender ratio of a role noun is often more relevant than the factual gender ratio, as behaviour is often driven by subjective experience. If perceived gender ratios for example influence career choices, factual ratios might change because of subjective perceptions. Even though there are certainly people who know the gender ratios for at least some roles (one might for example be aware of the gender ratio for one's own occupational role or have an occupational role for which this information is relevant, such as administrators in professional associations or in educational institutions), it is still sensible to assume that people have not memorised the exact factual gender

ratios of different roles. However, research by Garnham, Doehren and Gygax (2015), comparing the ratings of 290 of the English role nouns included in Misersky et al. (2014) with data from UK-governmental and academic sources, suggests that people are overall accurate at judging true gender ratios (even though with notable exceptions). This finding corroborates the claim that role nouns' gender associations as operationalised via perceived gender ratios are (at least in part) experience-based.

Against this background we collected perceived gender ratios in Chinese, Finnish, and Russian. We extend the established corpus of norms provided for Czech, English, French, German, Italian, Norwegian, and Slovak by Misersky et al., 2014, using the same rating scale, instructions, procedure, and online-tool as they did, hence facilitating cross-linguistic, cross-cultural, and longitudinal studies.

The presented languages are of particular interest as they represent languages that clearly differ from the mainly Germanic and Italic languages included in the norms by Misersky et al. (2014) by reference to the way in which they do, or do not, grammaticalize human referent gender. In Chinese human referent gender is not grammatically marked (Ettner, 2002). In Finnish most nouns but also pronouns are grammatically unspecified for human referent gender (Karlsson, 2018), whereas in Russian human referent gender is marked in nouns, pronouns, adjectives, and past-tense verbs (Corbett, 1982; Doleschal & Schmid, 2001).

For each language, the mean proportions of women judged to fill each role was calculated. Data are coded such that 0 indicated 0% women and 100% men, while 1 indicated 100% women and 0% men. These proportions (see By-item_results.pdf), together with the standard deviations, which indicate the level of consensus across participants in each sample, and the number of responses for each item, can be used to assess whether the terms are stereotyped, in the manner laid out by Misersky et al. (2014). As shown in Table 1, the ratings were highly reliable across the languages (correlations above the diagonal) and across scale directions (correlations on the diagonal).

| | CHINESE (*N* =80) | FINNISH (*N* = 77) | RUSSIAN (*N* = 135) |
|---|---|---|---|
| Chinese | .95* *420* | .83* *395* | .80* *418* |
| Finnish | | .98* *396* | .84* *395* |
| Russian | | | .97* *420* |

**Table 1** By-item analysis. Spearman's rank correlations (and *number of role nouns*) between the mean ratings per role noun between languages (above the diagonal) and between scale directions per language (on the diagonal).

* *p* < .001, *number of role nouns* differ due to different sized vocabularies.

The datasets contain all role nouns in English together with their translations into Chinese, Finnish, and Russian, the means and standard deviations per role nouns per language (by-item analyses) and the raw data.

# (2) METHODS

## 2.1 STUDY DESIGN

The online questionnaire from Misersky et al. (2014) was utilised. The questionnaire had been administered via a webpage hosted by the University of Fribourg (Switzerland; screen shots are available at https://dataverse.no/file.xhtml?persistentId=doi:10.18710/Y3P7BH/OBX7FE&version=2.0). After clicking on the link, participants were instructed to select their native language. Upon doing so, they were taken to the respective survey for their native language. These surveys each started with a welcome page in the participants' native language which provided information about the study, informed participants that their task was to rate the actual proportion of women and men in the groups presented by role nouns and required them to give informed consent through pressing a button labelled "Enter". Participants who pressed Enter reached the second page and were requested to enter demographic information (age, gender, first language, whether they are students and if so, at which University, what study program and what year of study). After this, participants rated a selection of 50% of the total role nouns (shown at a rate of 20 per page) for the proportion of women and men. These ratings were given on an 11-point scale that ranged from 0% to 100%, with intervals of 10%. For half of the participants the scale anchors were labelled "women 100/0" to "0/100 men", while for the other half the direction of the scale was "men 100/0" to "0/100 women" (see Figure 1). As in Misersky et al. (2014), participants were automatically divided into groups of four. For each group, the list of role nouns was randomly split in two. Role noun order was randomized per split, and either split was set up in both scale anchor versions. The four resulting sets were distributed to the four participants, such that the first and third and the second and fourth saw the same role nouns, and the first and second and the third and fourth were presented with the same scale anchor version. For every fifth participant this procedure was repeated. Within a group of four participants uncompleted datasets were discarded and the respective survey version repeated with a new

participant. Upon completing the survey, a code was provided such that student participants would then use that code as a verification of their completion of the study.

As outlined in Section 1, the results from the 11-point scale were encoded into the results files through the use of a proportional scale. Specifically, regardless of the items and scale direction an individual participant was presented with, each participant's response to each role they saw was recorded as a value between 0 (0% women, 100% men) and 1 (100% women, 0% men), with 0.5 representing the point of perceived gender balance (50% women, 50% men).

## 2.2 TIME OF DATA COLLECTION

Chinese data was collected between December 2018 and January 2019.

Finnish data was collected between October and December 2015.

Russian data was collected between December 2018 and February 2019.

## 2.3 LOCATION OF DATA COLLECTION

Chinese data was collected from students at Shandong Normal University, China.

Finnish data was collected from students at the University of Turku, Finland.

Russian data was collected from students at Tomsk State University, Russia.

## 2.4 SAMPLING, SAMPLE AND DATA COLLECTION

### Participants

***Overall sample.*** A total of 324 participants contributed to this study (Chinese = 81, Finnish = 81, Russian = 162). Data was gathered between October 2015 and June 2019. The data of 33 participants were removed because they were not native speakers of the target languages ($N$ = 10; Chinese = 1, Finnish = 1, Russian = 8), were not students ($N$ = 22; Chinese = 0, Finnish = 3, Russian = 19), did not comply with the instructions ($N$ = 0; Chinese = 0, Finnish = 0, Russian = 0), or were shown an incorrect file due to an internal system bug (N = 1; Chinese = 0, Finnish = 0, Russian = 1). The remaining 291 participants' datasets were used for further analysis.

***Chinese-speaking sample.*** Data in this set was gathered between December 2018 and January 2019. Following deselection, the dataset was composed of responses from 80 Chinese-speaking participants (61 female, 19 male), who were paid for their participation.

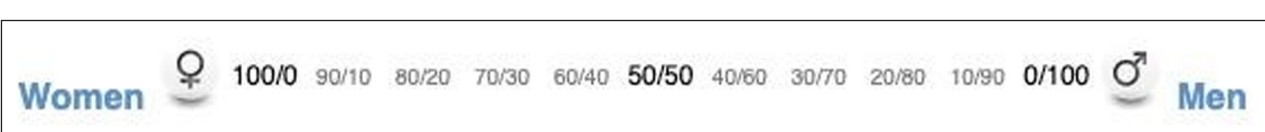

**Figure 1** Screenshot of scale anchors (English version, scale direction "women 100/0" to "0/100 men").

These participants were all psychology students from Shandong Normal University. Participants' ages ranged from 18 to 24 years old ($M = 20.44$, $SD = 1.11$). Participants were recruited among students.

***Finnish-speaking sample.*** Data in this set was gathered between October and December 2015. Following deselection, the dataset was composed of responses from 77 Finnish-speaking participants (69 female, 8 male), mainly from the University of Turku ($N = 74$). Participants received course credit for their participation. Participants' ages ranged from 18 to 50 years old ($M = 24.35$, $SD = 7.07$). Participants were recruited among the students taking introductory courses in psychology.

***Russian-speaking sample.*** Data in this set was gathered between December 2018 and February 2019. Following deselection, the dataset was composed of responses from 134 Russian-speaking participants (100 female, 32 male, 2 other/non-specified), mainly from Tomsk State University ($N = 124$). Participants received course credit as well as an appreciation certificate for their participation. Participants' ages ranged from 17 to 32 years old ($M = 20.07$, $SD = 2.55$). Participants were recruited among students.

## 2.5 MATERIALS/SURVEY INSTRUMENTS

The original list in English was composed of 422 role nouns, but the number of role nouns used in the survey varied across languages due to differences in vocabulary size (Chinese 420 role nouns; Finnish 396 role nouns; Russian 420 role nouns). The data set **Rolenounlist** provides a complete list of the role nouns in Chinese, Finnish, and Russian together with their English translation.

In Russian, a grammatical-gender language, grammatically marked role nouns were presented in both the masculine and feminine forms. The data set **Rolenounlist** presents the *feminine* form of these role nouns. In the survey, "(female)" and "(male)" was added to all role nouns in Russian and in Chinese. In Finnish this was only done for lexically gender marked role nouns (see the Footnotes in the dataset **Rolenounlist**). Role nouns were presented in the singular form for Chinese and in the plural form for Finnish and Russian. These minor variations are the result of language-specific peculiarities and correspond to what the researchers involved considered relevant to the intended future use of the collected norms.

## 2.6 QUALITY CONTROL

### Material Production

For all three languages examined in this dataset role nouns were translated from English into each respective language with the instructions (a) that the translation should match the English role noun as closely as possible, (b) to prefer gender unmarked forms to gender marked forms (e.g., postal carrier instead of postman – postwoman), and (c) to prefer native words to English

loan words. Role nouns were translated, and the translation was independently checked from a second translator. Duplicates and problematic items were identified, and disagreements were resolved between the two translators.

### Data screening

By-participant deselection and item-by-participant data screening was used. As outlined above, by-participant deselection occurred when participants were not native speakers of the target language (32 participants total), were not university students (10 participants total), or did not comply with the instructions (0 participants total). The first two requirements were tested through demographic questions (*What is your native language? Are you a university student?*), while the third requirement was tested through examining response patterns. Specifically, responses were examined to determine whether participants had adopted a response strategy that indicated they did not pay adequate attention to the instructions. This was operationalised as the removal of any participant who provided the same (or similar) responses to all items across the experiment.

## 2.7 DATA ANONYMISATION AND ETHICAL ISSUES

In this research, no personal data was processed. All participants gave their informed consent for inclusion through pressing a button labelled "Enter" before they participated in the study. Failure to pressing this button meant that the survey would not begin. IP addresses were recorded to prevent multiple entries by the same individual, but IP addresses were not associated to data nor were exported in the data file.

This study was conducted in accordance with the Declaration of Helsinki.

Research was conducted in line with the guidelines of the Finnish National Board on Research Integrity (https://tenk.fi/en/ethical-review/ethical-review-finland) and the Norwegian Committees for Research Ethics (https://www.forskningsetikk.no/en/). According to Finnish and Norwegian law, this research did not have to go through a process of ethical review in Finland and Norway. All participants gave written informed consent in accordance with the Declaration of Helsinki; the informed consent form was approved by the ethics committee of Shandong Normal University (China) and the ethics committee of Tomsk State University (Russia). Ethical approval for hosting the data collection was granted by the University of Fribourg (Switzerland).

## 2.8 EXISTING USE OF DATA

Kim, J.D. (2020). Connections between grammatical gender and occupational gender stereotypes: A thesis exploring the interplay between grammatical gender and occupational gender stereotypes as a method by

which wider knowledge related to the interplay between linguistic factors and stereotype beliefs can be expanded. Unpublished PhD thesis, NTNU, Trondheim, Norway.

# (3) DATASET DESCRIPTION AND ACCESS

The data set consists of a complete list of role nouns in the different languages (**Rolenounlist**), the by-item analyses (**By-Item**), the raw data sets (**RawData_CN, RawData_FI, RawData_RU**) and the code used to perform the by-item analyses (**Complete_Analysis**).

## 3.1 REPOSITORY LOCATION
https://dataverse.no/dataset.xhtml?persistentId=doi:10.18710/Y3P7BH

doi: 10.18710/Y3P7BH

## 3.2 FILE NAME; 3.3 DATA TYPE; 3.4 FORMAT; 3.5 LANGUAGE

| 3.2 FILE NAME | 3.3 DATA TYPE | 3.4 FORMAT | 3.5 LANGUAGE |
|---|---|---|---|
| Rolenounlists.pdf | Primary data | PDF/A | Chinese, English, Finnish, Russian |
| By-item_Results.pdf | Processed data | PDF/A | English |
| RawData_CN.csv | Primary data | CSV | Chinese, English |
| RawData_FI.csv | Primary data | CSV | English, Finnish |
| RawData_RU.csv | Primary data | CSV | English, Russian |
| CompleteAnalysis.rmd | R-code | RMD (R; R-Studio) | English |
| CompleteAnalysis.txt | R-code | TXT | English |

## 3.6 LICENSE
CC0

## 3.7 LIMITS TO SHARING
N/A

## 3.8 PUBLICATION DATE
17–August–2022

## 3.9 FAIR DATA/CODEBOOK
### Findability
Data is deposited in an open access repository at NTNU, and has been given a persistent identifier. This repository is searchable and discoverable online, and data hosted in this repository is given extensive metadata – the full names of all researchers (and contact details of primary author), the ability to add all relevant keywords, the ability to provide a detailed description of the dataset, the ability to link related publications, the depositor identity, the date of deposit, the language used to describe the data, any grant information, time period covered, dates of collection, kind of data, related material, related datasets, geospatial metadata (i.e., where in the world the data was collected),

and domain-specific metadata (i.e., the fields of research that were involved in the collection of the data).

### Accessibility
Data stored on the open access repository at NTNU is fully open access, and anyone wishing to see the files can retrieve them online following standard protocols.

### Interoperability
We have attempted to provide interoperability by using PDF, CSV and TXT files, as those are file types that can be opened by a broad range of different software solutions. The RMD file is potentially less interoperable as it requires the use of both R and RStudio but is set up so that a relatively new R user could start the program up, set the 'read' and 'write' commands to match where they have stored the files, press the 'run all' command, and have the data output for them without need for creating the analysis from scratch. Further, the code book below provides enough information that any individual wishing to analyse the dataset(s) through a different software solution should have suitable information to proceed.

### Reusability
We believe that the reusability of this dataset will be high. For the raw data files, the codebook provides a detailed explanation of the different parts, allowing the reader to perform alternative analyses (such as sub-sampling of the full dataset) if they so wish. For the processed data, the provided means and standard deviations can be used immediately as the basis for research using perceived gender ratios. For the complete analysis files, comments have been included to explain what each step of the code is doing; this not only provides readers with a greater understanding of, and confidence in, the results of this study, but provides them with far greater ability to replicate this research in further languages by knowing how to follow the same analytical path. Finally, clear open access licence and provenance information is included.

### Codebook for interpreting raw data
Each of the raw data files contains 434 columns and a varying number of rows depending on the number of participants in the sample.

The first row is a header row that contains the labels for each column, and the fourth row onwards each holds all responses given by a specific participant. Importantly, rows 2 and 3 contain the specific words shown on the left and right of the screen when a given occupation is shown; for example, if the role 'caretaker' is shown, then Chinese speakers would be presented with 看管员（女性）and 看管员（男性）, while Finnish participants would be presented with Talonmiehet (naiset) and Talonmiehet (miehet), and Russian participants would be presented with Сиделки (женщины) and Сиделки (мужчины). As such, analysis of the data should start from row 4.

                                                                                    

Columns 1 through 10 contain demographics information, while columns 11 through 432 contain occupational ratings, and finally columns 433 and 434 were used for ensuring all roles were presented equally.

The column headed 'ID' contains an identifying number per participant to allow for by-participant deselection. Rows 2 and 3 instead contain the words SetA and SetB respectively; the inclusion of these labels allow for the automatic removal of these rows using analytical software such as R.

The column headed 'language' indicates the native language of each participant. For replication of the results presented in the By-item_Results.pdf, non-native speakers should be removed.

The column headed 'gender' contains the participants' reported gender; F for female, M for male, and O for other.

The column headed 'institution' indicates which higher-education institution they are affiliated with.

The column headed 'studentstatus' indicates whether each participant was or was not a university student when responding to the experiment; Y if they were a student, N if they were not a student. For replication of the results presented in the By-item_Results.pdf, non-students should be removed.

The column headed 'year of study' is intended to capture where exactly in their higher-education studies each participant was. However, some participants instead interpreted this as asking for the year in which they started their degree, or the year in which they undertook the study, either in full (e.g., '2015') or in short (e.g., '18'). On a per-language basis, there were 4 Chinese (1 who listed their starting year, 3 who listed the 'current' year), 0 Finnish, and 38 Russian (18 who listed their starting year, 20 who listed the 'current' year) participants who produced this error. This does not entirely preclude the use of 'year of study as being useful for further analysis, however; since it is unlikely for students to take multiple decades or longer to complete their studies, and seeing as we know that all results in these datasets were gathered between 2015 and 2019, then 1) recoding and 2) by-participant deselection measures could be introduced for participants whose 'year of study' number was equal to or greater than 15. For this, a comparison should be made between the year listed in the 'test date' column (described below) and the year in the 'year of study' column. For example, one could automatically do this for these participants by subtracting the 'year of study' number in full form (i.e., two responses from Chinese participants [ID: 3305, 3308] and one Russian participant [ID: 3432] would need to be recoded from short to full form first) from the year listed in the 'test date' column and printing the results to a new column; a 0 in the new column would therefore be indicative of the participant giving the current year, and thus being eligible for by-participant deselection, while any number above 0 would be indicative of the participant giving their starting year, and can therefore be recoded into the 'year of study' column.

The column headed 'description' is intended to capture, for participants who indicated that they were not students, what exactly their occupation was.

The column headed 'test date' indicates the specific day that each participant undertook the experiment on.

Columns 11 through 432 each refer to a specific occupation about which the participant was asked. In each column, the numeric value indicates the gender ratio that they associate with that occupation, with 0 indicating 100% men and 0% women, 1 indicating 100% women and 0% men, and 0.5 indicating 50% women and 50% men. Not every occupation was shown to every participant; rows where a column is empty indicates that it was not presented to that participant. Further, not every occupation was shown for every language; this is observable in the raw data files as columns in the 11 through 432 range, aside from the first three rows, are empty. This indicates that the occupation was not presented at all to participants. This was either because multiple roles in English translated into the same role in that language, or because there were no appropriate translations for the role. Explanations for each role per language can be seen in Rolenounlists.pdf file. The reason for why these roles are included in the raw data file is that it standardises the results files, minimising the steps needed to compare results per language.

Column 434, headed 'Type', indicates two things; firstly, which occupations a given participant will be shown, and secondly, whether they would see the male or female symbol (and, where appropriate, grammatical or lexical gender referent) on the left of the screen. The second is simply indicated by the two words; i.e., 'left fem' means the female symbol/referent was on the left, while 'left mas' means the male symbol/referent was on the left. In keeping with the explanation of columns 11 through 432, responses were encoded into the raw data files such that, regardless of whether the female or male symbol/referent was on the left, 0 indicated 100% men and 0% women, while 1 indicated 100% women and 0% men. The first is indicated by the number; different four-part 'sets' were created, and can be understood in combination with Column 433, 'Session', which lists each number four times; the rows that have a shared Session number are part of the same 'set'. The sets were designed such that the roles were split evenly into two lists that covered all roles without repetition, and that, within the set, participants for whom column 434 contained a 0 or 2 would respond to one set of roles, while participants for whom column 434 contained a 1 or 3 would respond to the other set of roles. Further, it was intended that participants for whom column 434 contained a 0 or 1 would see the male symbol/referent

on the left, while participants for whom column 434 contained a 2 or 3 would see the female symbol/referent on the left. This was indeed the case for the Chinese and Russian samples, ensuring that, within each set, each role was presented to a participant with both female left and male left. However, for the Finnish sample, an error occurred in which participants for whom column 434 contained a 0 or 2 saw the male symbol/referent on the left, while participants for whom column 434 contained a 1 or 3 saw the female symbol/referent on the left. This meant that, within each set, each role was presented twice with the same gender symbol/referent on the left. While this does present an issue, an examination of the results for Finnish split by gender direction indicated that the number of responses per role were relatively similar (female left: mean = 18 responses, SD = 6 responses; male left: mean = 18 responses, SD = 6 responses), while Spearman correlational analysis indicated a correlation of 0.98 between female and male left. Finally, results within a given set were not always presented in a neat sequential order; for example, in the Finnish dataset, the first three participants who were a part of Session 464 appear on three consecutive lines, with the final participant in set 464 appearing eight rows later.

## (4) REUSE POTENTIAL

The data sets provide social scientists with gender ratio information when selecting social or occupational roles to systematically vary or control for stimulus material's gender associations. Researchers typically collect such information as part of the preparations towards an experiment. As outlined by Misersky et al. (2014), different approaches might be chosen, especially regarding the instructions and how the information is being measured (e.g., rating scales). This heterogeneity makes it difficult to compare between and within languages. To remedy such challenges, we utilised the questionnaire provided by Misersky et al. (2014) and followed their procedures for data collection, data preparation and analysis. Our data sets hence expand their data sets, which are regularly cited, and can be used when investigating cultural variation and historic change in cross-linguistic or cross-cultural perspective.

## ACKNOWLEDGEMENTS

We thank Changze Yhao, Yingliang Zhang, Jiu-Ju Su, Pirita Pyykkönen-Klauck, Sari Forsblom, Alina Vasilieva and Ksenia Pozovkina for their support on translating the role noun lists as well as Maurizio Rigamonti for his support on maintaining the questionnaire website.

## FUNDING STATEMENT

This research was partially funded by:

Nordic-Russian Cooperation Programme in Education and Research, project number NCM-RU-2015/10045.

The Research Council of Norway, project number 240881.

## COMPETING INTERESTS

The authors have no competing interests to declare.

## AUTHOR CONTRIBUTIONS

**Ute Gabriel**, Norwegian University of Science and Technology, Trondheim, Norway, Funding acquisition, Methodology, Project administration, Resources, Validation, Writing Original draft, Review and Editing.

**Jonathan Kim**, Norwegian University of Science and Technology, Trondheim, Norway, Data curation, Formal analysis, Writing Original draft, Review and Editing.

**Anton Oettl**, Norwegian University of Science and Technology, Trondheim, Norway, Project administration, Supervision, Review and Editing.

**Pascal Gygax**, University of Fribourg, Switzerland, Funding acquisition, Methodology, Resources.

**Lei Cui**, Shandong Normal University, China, Investigation, Resources, Validation, Review and Editing.

**Jukka Hyönä**, University of Turku, Finland, Funding acquisition, Investigation, Resources, Validation, Review and Editing.

**Olga Nagel**, Tomsk State University, Russia, Funding acquisition, Investigation, Resources, Validation, Review and Editing.

## AUTHOR AFFILIATIONS

**Ute Gabriel** orcid.org/0000-0001-6360-4969
Norwegian University of Science and Technology, NO

**Jonathan Kim** orcid.org/0000-0002-7926-6834
Norwegian University of Science and Technology, NO

**Anton Öttl** orcid.org/0000-0002-5711-4362
Norwegian University of Science and Technology, NO

**Pascal Gygax** orcid.org/0000-0003-4151-8255
University of Fribourg, CH

**Lei Cui** orcid.org/0000-0003-3584-8477
Shandong Normal University, CN

**Jukka Hyönä** orcid.org/0000-0002-5950-3361
University of Turku, FI

**Olga Nagel** orcid.org/0000-0001-6210-1526
Tomsk State University, RU

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

## PEER REVIEW COMMENTS

*Journal of Open Psychology Data* has blind peer review, which is unblinded upon article acceptance. The editorial history of this article can be downloaded here:

- **PR File 1.** Peer Review History. DOI: https://doi.org/10.5334/jopd.73.pr1

Gabriel et al. *Journal of Open Psychology Data* DOI: 10.5334/jopd.73

**TO CITE THIS ARTICLE:**
Gabriel, U., Kim, J., Öttl, A., Gygax, P., Cui, L., Hyönä, J., & Nagel, O. (2023). Norms on the Gender Perception of Role Nouns: Gender Ratio Data for Chinese, Finnish, and Russian. *Journal of Open Psychology Data,* 11: 2, pp. 1–9. DOI: https://doi.org/10.5334/jopd.73

*Journal of Open Psychology Data* is a peer-reviewed open access journal published by Ubiquity Press.

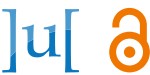