## [Peer Review History. · Journal of Open Psychology Data]

Peer Feedback on "Norms on the gender perception of role nouns: Gender ratio data for Chinese, Finnish, and Russian"

Dear Dr Ute Gabriel,

Thanks for submitting your work for review, and for your patience whilst we've sought out reviewers. We had some difficulties attempting to find someone suitable to co-review with a student who very kindly volunteered to review for JOPD, but I am now pleased to confirm that this process has now been completed and we have two positive reviews of your work. After review, we have reached a decision regarding your submission to Journal of Open Psychology Data, "Norms on the gender perception of role nouns: Gender ratio data for Chinese, Finnish, and Russian ". Our decision is to request minor revisions of the manuscript prior to acceptance for publication.

The full review information is included at the bottom of this email. Please note that there may also be a copy of the manuscript file with reviewer comments available once you have accessed the submission account. We ask you to please consider the issues and revise the file accordingly, as you see fit. Please make all changes in 'tracked changes' or different coloured text to speed up the next part of the process. There really are very minor comments so this 'revisions' decision is an opportunity to revise your work in-line with the feedback and to go through things carefully before further processing.

Instructions for how to resubmit your article online are pasted below. Please ensure that your revised files adhere to our author guidelines, and that the files are fully proofed prior to upload. Please also include a revised version of your article with 'tracked changes', adding comments where appropriate, to indicate the revisions made, in addition to a brief document outlining how you have responded to the reviewers' requests.

Please also ensure that all copyright permissions have been attained for any figures/tables you have included.

If you have any questions or difficulties during this process, please do contact us.

The scope of changes is minimal, so I imagine this won't take long but do keep me posted and we'll look forward to actioning the revision shortly.

Kind regards,

Dr Thomas Rhys Evans

University of Greenwich

Thomas.Evans@greenwich.ac.uk

Reviewer A:

The paper titled “Norms on the gender perception of role nouns: Gender ratio data for Chinese, Finnish, and Russian” addresses an interesting and relevant question and provides an interesting data set. As far as I can tell, all criteria are met. Overall, it seems to me to be an exemplary paper and I recommend its publication.

The methods section is described in such detail that it seems easy to replicate the study.

The dataset is deposited in a repository where it can be found permanently and easily. The data are provided with a CC0 license and the file formats are all readable with free and open access software. The datasets are well described, making it easy to follow. I can imagine the data being interesting for reanalysis and comparison with other datasets. The dataset complies with the FAIR guidelines.

The authors state that the study was in accordance with the Declaration of Helsinki, that the subjects consented to the study, how the project was funded, that they have no conflict of interest, and who contributed what.

I have not installed RStudio. Therefore, I only checked the Complete_Analysis.txt file. I spot checked some results and was able to replicate them without any problems.

It may be due to the version I read, but I did not find any information about the authors' ORCID in the article. These might be added.

Reviewer H:

For this review, I primarily focused on the journal peer review sections as well as the data paper template for JOPD to ensure each section had sufficient detail. Overall, the paper is excellent, expands on prior literature, and gives useful insight into a poignant topic; I only have minor comments. I was able to load and examine the dataset in R and am confident the dataset is reusable. While I am not an expert on the topic, the papers' background section gave me a good understanding. Sections 2 and 3 elucidated how the study was conducted and the additional documentation was helpful. Here are some minor comments:

1. The paper contents

- a. The methods section of the paper must provide sufficient detail so that a reader can understand how the dataset was created, and would within reason be able to recreate it.

A brief comment on how the random allocation and role noun splitting occurred would be beneficial (Section 2.1).

The coding of datasets (0-1) should be briefly commented on in the method section to allow for meaningful secondary data use (as stated in section 2 in the data paper template for JOPD). While the coding is mentioned in sections 1 and 3, looking at the method section independently provided minor confusion as to the coding of variables, e.g., "80"/"20".

The link <https://www4.unifr.ch/lcg> is seemingly no longer active – it cannot be reached. The full questionnaire could be provided if possible. I can imagine how I would recreate the questionnaire, however, providing the questionnaire would allow for further reproducibility/replication. If this is not possible, a statement on the copyright of materials, as it is based on the Misersky et al. (2014) questionnaire, could be provided (as stated in section 2.5 of the data paper template for JOPD).

b. the dataset must be correctly described.

This includes where participants misinterpreted questions, e.g., years of study. A brief comment on potential steps researchers could take to use "years of study" as a variable for analysis might prove beneficial.

2. The deposited data

f. studies involving human subjects should adhere to local ethical standards at the host institution and follow the American Psychological Association's (APA) Ethical Principles of Psychologists and Code of Conduct (<http://www.apa.org/ethics/code/index.aspx>). Participant data should be sufficiently anonymized and appropriate consent forms should be signed.

The data is sufficiently anonymous. However, section 2.7 should include any institutional review boards that granted approval for the study considering it was a multi-nation study.

Journal of Open Psychology Data

<http://openpsychologydata.metajnl.com>